# Unsupervised learning of haptic material properties

Anna Metzger[1,2†], Matteo Toscani[1,2*†]

[1]Department of Psychology, Bournemouth University, Bournemouth, United Kingdom; [2]Department of Psychology, Justus-Liebig University, Giessen, Germany

**Abstract** When touching the surface of an object, its spatial structure translates into a vibration on the skin. The perceptual system evolved to translate this pattern into a representation that allows to distinguish between different materials. Here, we show that perceptual haptic representation of materials emerges from efficient encoding of vibratory patterns elicited by the interaction with materials. We trained a deep neural network with unsupervised learning (Autoencoder) to reconstruct vibratory patterns elicited by human haptic exploration of different materials. The learned compressed representation (i.e., latent space) allows for classification of material categories (i.e., plastic, stone, wood, fabric, leather/wool, paper, and metal). More importantly, classification performance is higher with perceptual category labels as compared to ground truth ones, and distances between categories in the latent space resemble perceptual distances, suggesting a similar coding. Crucially, the classification performance and the similarity between the perceptual and the latent space decrease with decreasing compression level. We could further show that the temporal tuning of the emergent latent dimensions is similar to properties of human tactile receptors.

## Editor's evaluation

The authors apply machine learning techniques to empirical-based simulations of the vibrotactile patterns to find their compressed representations. These compressed representations turn out to be similar to those deduced from human responses and also correspond to those expected based on efficiency arguments.

*For correspondence:
mtoscani@bournemouth.ac.uk

†These authors contributed equally to this work

Competing interest: The authors declare that no competing interests exist.

## Introduction

With our sense of touch, we are able to discriminate a vast number of materials. We usually slide the hand over the material's surface to perceive its texture (*Lederman and Klatzky, 1987*). Motion between the hand and the material's surface elicits vibrations on the skin, which are the sensorial input mediating texture perception (*Weber et al., 2013*).

It was proposed that perceptual representations emerge from learning to efficiently encode sensorial input (*Barlow, 1961*; *Olshausen and Field, 1996*; see for a review *Simoncelli and Olshausen, 2001*). For example, in color vision, the excitations of long- and middle-wavelength-sensitive cones are highly correlated. At the second stage of processing, still in the retina, a transformation into two-color opponent and a luminance channel achieves an efficient and decorrelated representation, akin to a principal components analysis (PCA of the input signals *Buchsbaum and Gottschalk, 1983*; *Zaidi, 1997*; *Gegenfurtner, 2003*). Receptive field properties in the early visual pathway (*Atick and Redlich, 1992*; *Olshausen and Field, 1996*) as well as the tuning properties of auditory nerve fibers (*Lewicki, 2002*; *Smith and Lewicki, 2006*) can emerge by efficiently encoding natural images or sounds, respectively. Recently, it was shown that efficient coding could also explain the simultaneous development of vergence and accommodation as a result of maximizing coding efficiency of

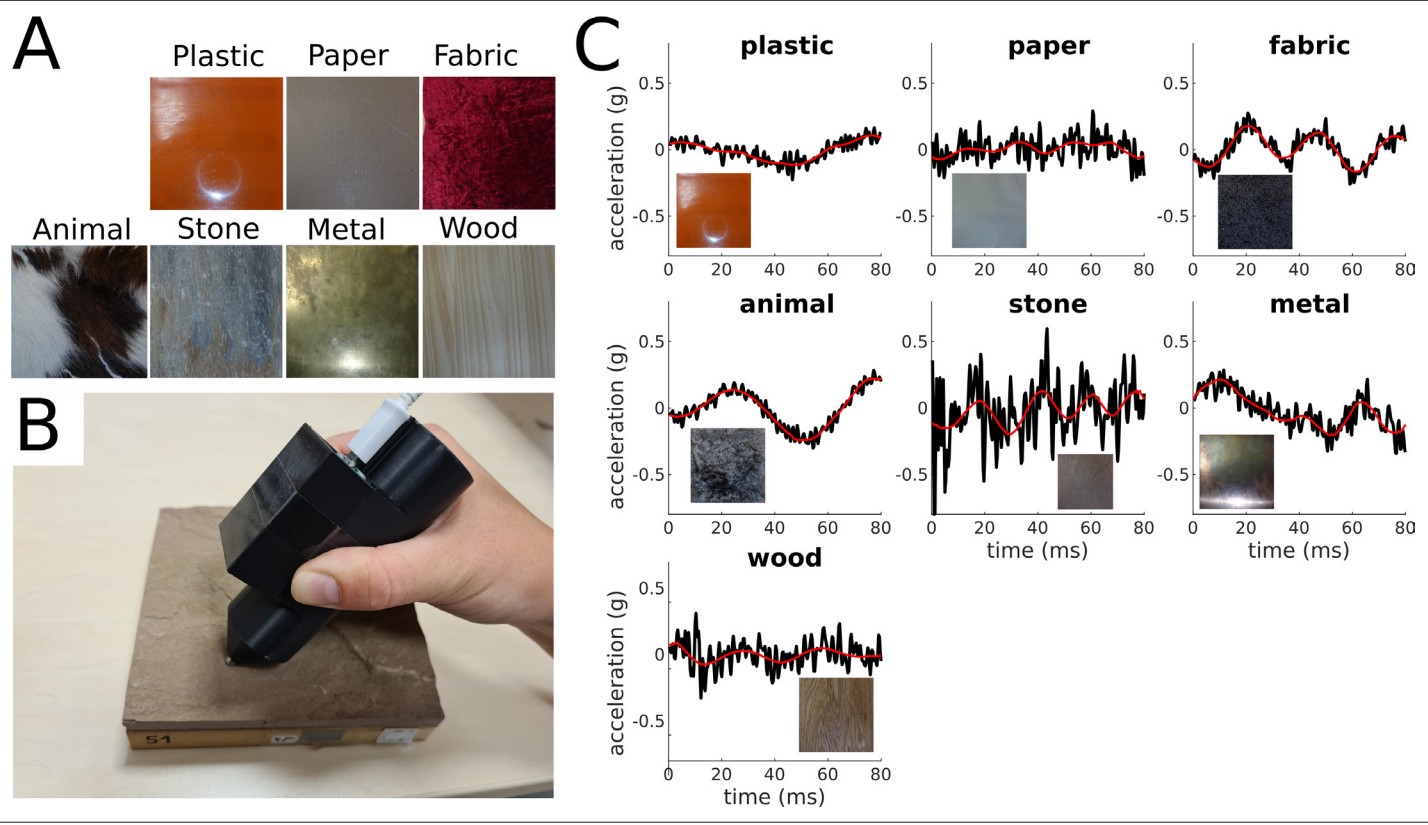

**Figure 1.** Materials and vibratory signals. (**A**) Examples of materials. One example per category is shown. (**B**) Accelerometer mounted on a 3D-printed pen with a steel tip used for acquisition of vibratory signals. (**C**) Original and reconstructed acceleration traces for one example material per category. The black lines represent the original amplitude signals (y-axis) over time (x-axis), and the red line represents the corresponding reconstructed signals.

the retinal signals (*Eckmann et al., 2020*). There is currently a lot of interest whether higher-level representations can also be learned by efficient encoding of the retinal images (*Fleming and Storrs, 2019*).

Here, we explore whether the perceptual representations of our haptic world can be learned by encoding vibratory signals elicited by the interaction with textures. We used a Deep Convolutional Autoencoder, a framework for unsupervised learning, to reconstruct acceleration recordings from free explorations of 81 everyday materials (*Figure 1A and C*).

We trained the Deep Convolutional Autoencoder to reconstruct the vibratory signals after compressing them into a 16-features latent representation. Then, we related the latent representation to haptic material perception: we show that, based on the emerging latent representation, it is possible to classify the vibratory signals into the different material categories assigned by human participants. We computed the centroids of each category within the latent space and showed that the distances between these categories resemble perceptual distances, measured with a rating experiment. These results suggest that the latent representation produced by unsupervised learning is similar to the information coding of the haptic perceptual system.

To interpret the dimensions of the latent space, we mimicked a physiology experiment. We generated a large number of sinusoids systematically varying in frequency and computed the corresponding representation within the latent space. We observed that the temporal tuning of the latent dimensions is similar to properties of human tactile receptors responsible for perception of haptic textures (i.e., Pacinian [PC] and rapidly adapting [RA] afferents).

As a control, we repeated all analyses after reducing the compression level by increasing the number of latent dimensions. Crucially, the similarity between the latent representation and perceptual representation increases with compression. This suggests that perceptual representations emerge by efficient encoding of the sensory input signals.

# Results

## Behavior

Participants were instructed to slide the steel tip of the pen containing the accelerometer over the surface of the material sample (*Figure 1B*). Visual and audio signals were excluded. After a 10-s-long exploration, participants rated each material based on seven descriptors of its haptic properties (roughness, orderliness, hardness, temperature, elasticity, friction, and texture). We used the same materials and similar descriptors as previously used by *Baumgartner et al., 2013*, who investigated haptic and visual representations of materials. Crucially, in their study participants could touch the stimuli with their barehands; therefore, temperature and softness information was available, whereas in our study participants had mostly to rely on vibratory signals for rating the materials.

To visualize the perceptual representation emerging from the ratings, we performed a PCA on the ratings averaged across participants and plotted each material in the space defined by the first two principal components (*Figure 2A*). This is helpful because ratings based on different descriptors might be highly correlated; thus, the main principal components offer a compact representation of the responses. In fact, the first two principal components explain 65.9% of the total variance, very close to the 70.8% found by *Baumgartner et al., 2013*.

The emerging representation is remarkably similar to the one discovered by *Baumgartner et al., 2013*, despite no temperature and limited softness information (*Figure 2B*). To quantitatively asses this similarity, we computed distance matrices from the ratings (*Figure 2C and D*). To do so, we correlated the ratings for each pair of materials, then averaged the obtained correlations, resulting in a correlation value for each pair of categories, which is a measure of how close the two categories were perceived. Finally, correlations were transformed into perceptual distances with the following equation:

$$\delta = \sqrt{2(1 - \rho)} \tag{1}$$

The distance matrix we obtained is highly correlated with the one from *Baumgartner et al., 2013*: $r = 0.69$, $p < 0.05$, indicating that perceptual representations are very similar. The missing temperature and limited softness information did not much affect the perceptual representation we inferred from our rating experiment, which could therefore be thought as representative for natural haptic perception.

## Unsupervised learning

The Autoencoder learned a compressed latent representation to reconstruct the signals provided as input (*Figure 1C*). We evaluated reconstruction performance by correlating the input with the corresponding output signals and computing $R^2$. Performance is computed for the train set (95% of the data) and the test set (remaining 5%). The reconstructed signals could explain 25% of the variance of the original signals for the train set and 23% for the test set. The learned latent representation is redundant: 95% of the variance of the latent representation of the full data set can be explained with 10 principal components.

We used the t-distributed stochastic neighbor embedding (t-SNE) method (*Hinton, 2008*), a nonlinear dimensionality reduction technique for visualization, to represent the 10D *latent PCs space* in a 2D space (*Figure 3A*). This seems a faithful representation of the *10D latent PCs space* as the distances between the category averages in the 2D t-SNE space are highly correlated with the ones computed in the 10D space ($r = 0.994$). Some categories were easy to discriminate, some more difficult. This is visible from the category averages. For instance, *metal* is far apart from *animal* but close to *wood*.

Linear classification within the 10D *latent PCs space* could identify the correct ground truth category better than chance (i.e., 36%; empirical chance level given by a bootstrap analysis = 14%, with [11–22%] 95% confidence interval). However, samples of one material might be systematically perceived as being made of a different material. Hence, we also asked participants after the exploration of each material how much it 'felt like' each of the seven categories. This way we could assign perceived categories to each material sample based on the participants' ratings. For each participant, we assigned for every material sample the category label corresponding to the highest rated category, and then we assigned for each material sample the most frequent label among participants (i.e., majority vote). Crucially, classification performance improved when the linear classifier was used

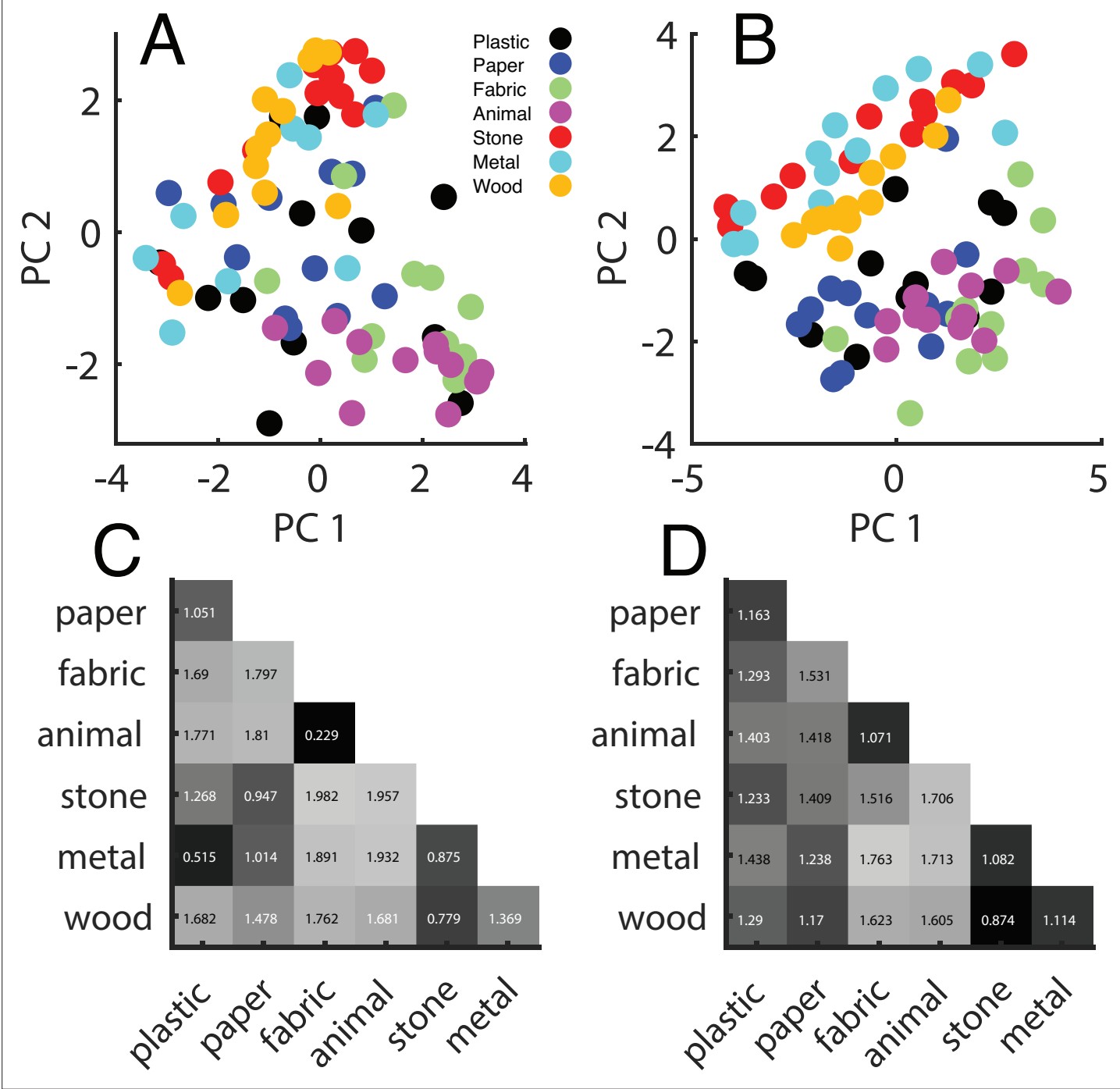

**Figure 2.** Perceptual ratings. (**A**) Perceptual representation within the first (x-axis) and the second (y-axis) principal components space. Principal components analysis (PCA) was performed on the z-transformed ratings pooled over participants. Each point represents one material, and different colors represent different categories. (**B**) Same representation as in (**A**) based on barehand explorations (adapted from Figure 5 of *Baumgartner et al., 2013*). (**C**) Distance matrix based on the ratings from our experiment and (**D**) based on *Baumgartner et al., 2013* data.

to predict the labels given by human participants (40%, empirical chance level given by a bootstrap analysis = 14%, with [11–22%] 95% confidence interval). This classification performance is very close to the between-participants agreement level (44%), indicating that the information present in the *latent space* is almost enough to explain human category judgments.

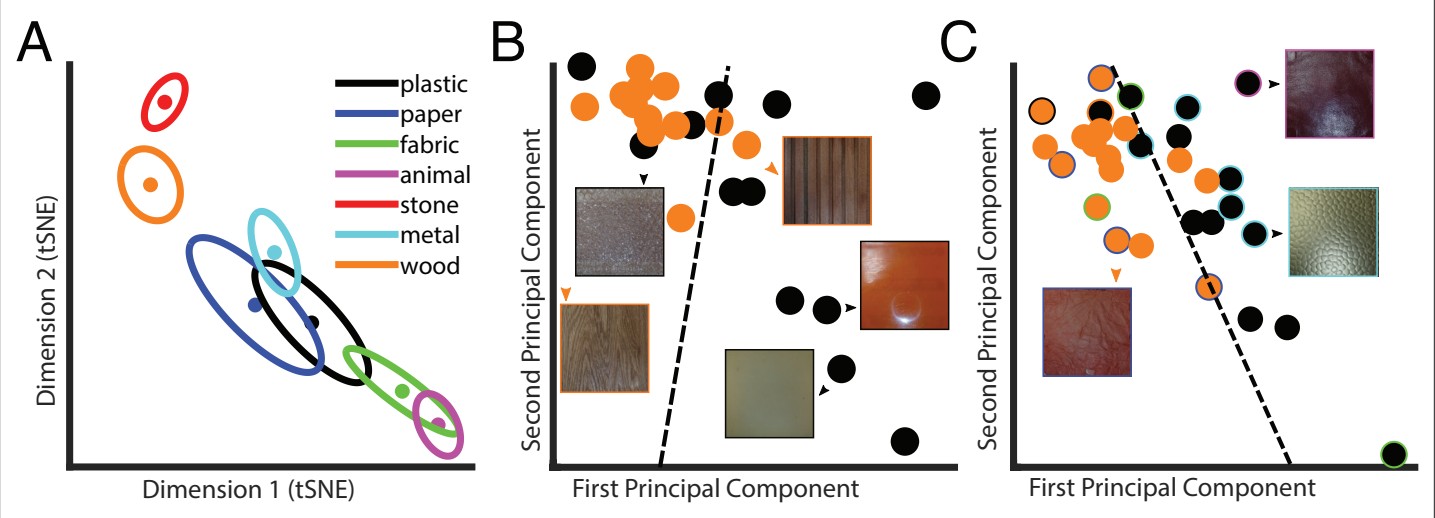

**Figure 3.** Latent space representation of the Autoencoder. (**A**) Centers of categories in the t-distributed stochastic neighbor embedding (t-SNE). Each color represents a different category, as indicated in the legend. Dots represent material categories averaged across material samples, with the two-dimensional errors represented by error ellipses (axes length given by 1/2 of the eigenvalues of the covariance matrix; axes directions correspond to the eigenvectors). (**B**) Example of material category classification in the two first principal components of the latent PCs space, based on the ground truth material category labels. The black data points are the 2D representation of plastic; wood is in orange. The icons represent examples of material samples. The dashed line is the classification line learned by a linear classifier, which could tell apart the two categories with 75% accuracy. (**C**) Same as in (**B**), but based on perceptual labels. Black and orange data points indicate the perceptual categories plastic and wood, respectively. The data points surrounded by a colored rim represent material samples that were perceived as wood or plastic but belong to a different ground truth category (indicated by the color of the rim with the same color code as in **A**). The dashed line is the classification line learned by a linear classifier, which could tell apart the two categories with 87% accuracy. Note that the classification results reported in the main text are based on 10 principal components.

Classification based on the raw vibratory signals could be achieved only by extracting known features related to the psychophysical dimensions of tactile perception such as hardness or friction (*Strese et al., 2017*).

We used the distances between centers in the latent PCs space to compute a distance matrix (*Figure 4A*), which is remarkably similar to the one obtained from the behavioral results. Crucially, the distances between centers of categories in the *latent PCs space* correlate with perceptual similarity (*Figure 4B*, black data points; N = 21, r = 0.74, p<0.0005), indicating that the representation of material properties within the compressed space learned by the Autoencoder from vibratory signals resembles the haptic perceptual representation.

*Baumgartner et al., 2013* found similar perceptual representations of materials with purely visual and purely haptic stimulations. We repeated the correlation analysis based on the distance matrix computed from the rating data from *Baumgartner et al., 2013* for the vision-only and the touch-only conditions. Again, perceptual distances based on the ratings in the touch-only condition correlate with the distances between centers of categories in the *latent PCs space* (*Figure 4C*, black data points; N = 21, r = 0.72, p<0.05). This is consistent with the similarity between the distance matrices from our and their rating experiment. However, the same analysis repeated based on the ratings from the vision-only condition yields similar correlation results (*Figure 4C*, gray data points; N = 21, r = 0.66, p<0.05). This is intriguing because visual and tactile signals originating from the same material are fundamentally different. However, similar physical properties are likely to create similar patterns of covariation in the sensory inputs. The similarity between the visual and the haptic material representations, and the representation emerging from unsupervised learning corroborates the idea that by learning to efficiently represent the sensory input perceptual systems discover the latent variables responsible for the sensorial input (*Fleming and Storrs, 2019*), that is, the systematic differences between materials.

Our results showed a remarkable similarity between perceptual representations and how material categories are represented in the latent space learned by the Autoencoder. This is consistent with the theory that perceptual representations emerge from efficient encoding of the sensory signals. Specifically, we believe that by compressing information the Autoencoder has learned the regularities due

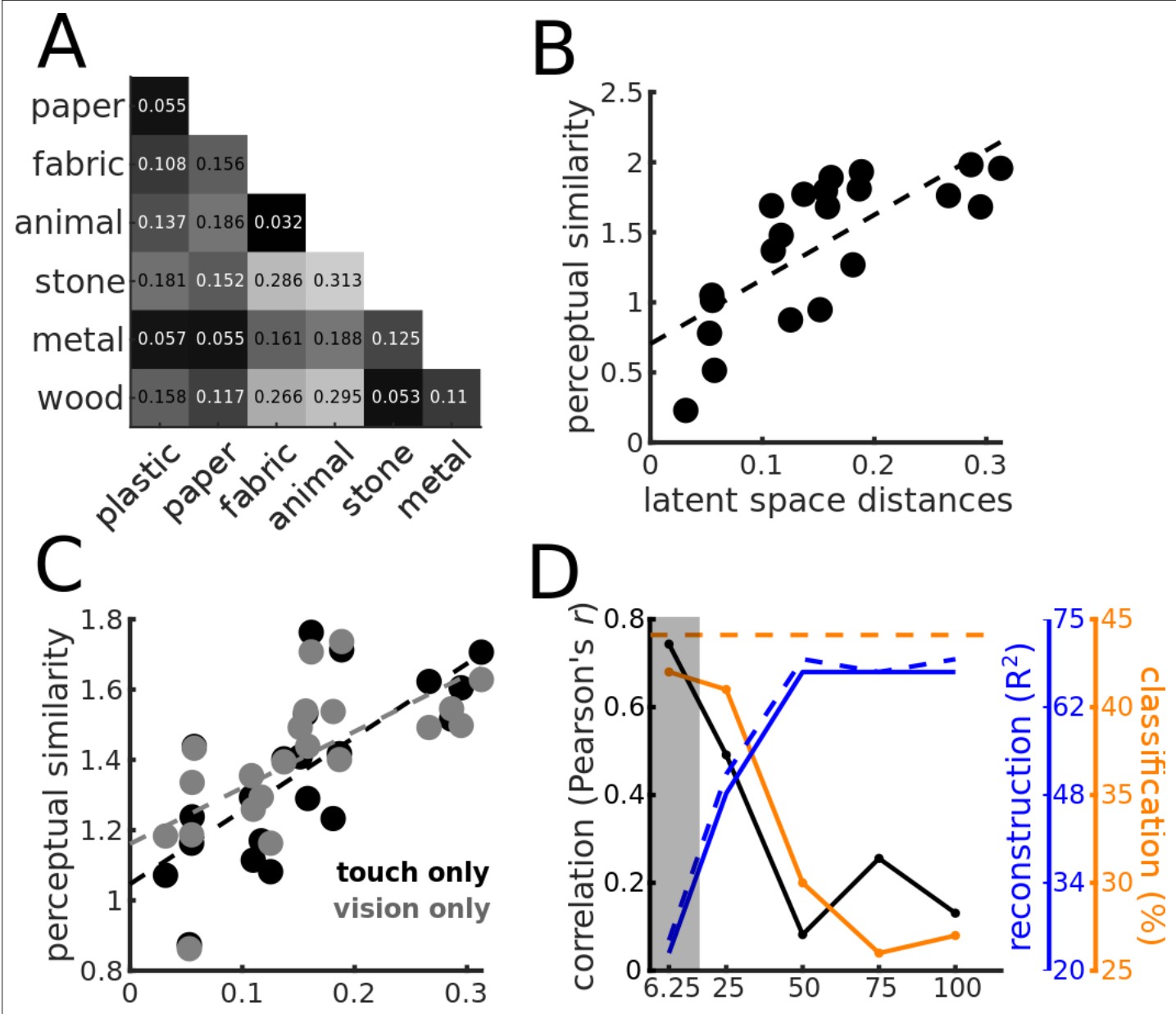

**Figure 4.** Similarity between the perceptual and the latent representation of the Autoencoder. (**A**) Distance matrix based on the distances between the category centers in the (10D) latent PCs space. (**B**) Relationship between perceptual distances (y-axis) and distances between categories in the latent PCs space (x-axis). The black dashed line indicates the linear regression line. (**C**) Same as (**B**), but based on the rating experiments from **Baumgartner et al., 2013**, for the touch-only (black data points) and vision-only (gray data points) conditions. Again, the dashed lines represent regression lines, and in black and gray for the touch-only and the vision-only conditions, respectively. (**D**) Effect of compression. Correlation between distances in the latent PCs space and perceptual distances (black y-axis), reconstruction performance (blue y-axis), and classification performance (orange y-axis), as a function of compression rate (x-axis). Black line and data points indicate Pearson's r correlation coefficient, and orange line and data points indicate classification accuracy (%). The blue lines represent the reconstruction performance computed as $R^2$ and expressed in percentage of explained variance; the dashed blue line represents the reconstruction performance for the training data set, and the continuous line for the validation data set. The horizontal orange dashed line represents the between-participants agreement expressed as classification accuracy. The gray shaded area indicates the compression level corresponding to the Autoencoder used for the analyses depicted in **Figure 3** and **Figure 4A–C**.

to the physical properties of the different materials and discounted the variability due to the specific properties of each sample, e.g., it has learned to represent the general properties of wood rather than the specific properties of each wooden sample. We argue that such generalization process resembles how we build perceptual representations, that is, compression is responsible for the similarity between

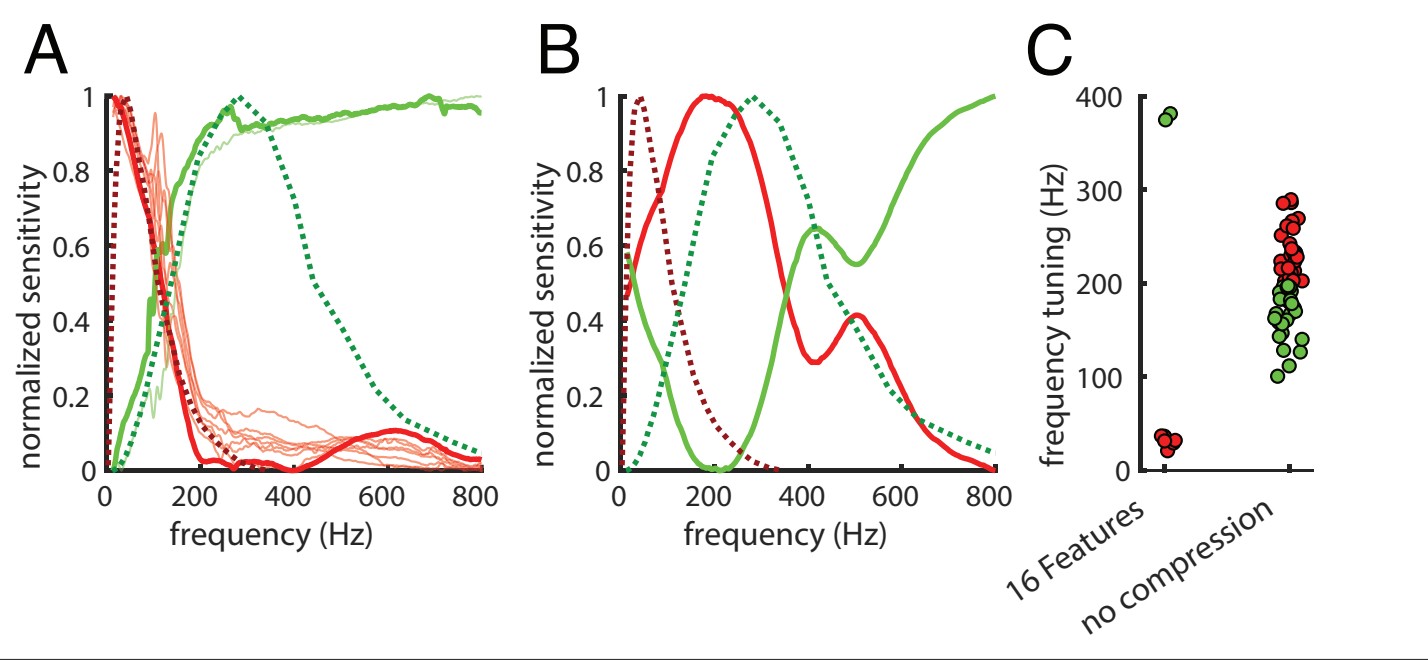

**Figure 5.** Frequency tuning.
(**A**) Comparison between temporal tuning of the dimensions of the 10D latent PCs space and the temporal tuning of the Pacinian (PC) and rapidly adapting (RA) afferents (green and red dashed lines, respectively). Tuning is given by sensitivity (y-axis) as a function of temporal frequency (x-axis). RA is optimally tuned at 40–60 Hz, and the PC is optimally tuned at 250–300 Hz. The first two principal components of the latent PCs space are represented by thick lines; thinner lines represent the other components. For a group of components (in red), sensitivity peaks around the RA afferents peak, the other group (in green) reaches the maximum sensitivity in proximity of the PC peak. (**B**) same as (**A**), but for the Autoencoder with no compression (100% compression rate). Here, only the first two principal components of the latent PCs space are shown as the tuning profiles tend to differ from each other. (**C**) Frequency tuning for the 16 features (6.25% compression rate) and the no compression (100% compression rate) Autoencoder. Tuning is again defined as average frequency weighted by sensitivity, and colors are assigned by the *k*-means clustering algorithm.

perceptual representations and representations within the latent space. We tested this prediction by gradually decreasing the compression rate of the Autoencoder until no compression is applied, that is, the dimensionality of the latent space is the same as the one of the input. Higher compression rate, defined as the ratio between the dimensionality of the latent space and the one of the input (in percentage), implies lower compression and results in higher reconstruction performance (*Figure 4D*, blue continuous and dashed lines, for the validation and the training data set, respectively). The least the compression (higher compression rate), the lower the correlation between perceptual and latent space distance matrices (*Figure 4D*, black), and the lower the classification performance (*Figure 4D*, orange). These results show that compression is crucial for the Autoencoder to learn an internal representation that resembles perceptual representations, that is, perceptual representations emerge by efficiently encoding the sensory input. In fact, because of the bottleneck, Autoencoders must learn a compressed representation from which the input signals can still be reconstructed. To do so, they tend to discover latent variables that are good at capturing the pattern of covariation in the input generated by distal causes (e.g., material properties of the surfaces, *Fleming and Storrs, 2019*; *Storrs et al., 2021*).

Haptic textures are mainly sensed by the PC and RA afferents, which are highly sensitive to skin vibrations (*Weber et al., 2013*) and have known temporal tuning properties (*Kandel et al., 2000*; *Mountcastle et al., 1972*). We probed the model's temporal tuning by mimicking a physiology experiment: we presented the network with a large number of sinusoids systematically varying in frequency within the perceptually relevant range (*Weber et al., 2013*). We defined the temporal tuning of the *latent PCs space* as the responses to sinusoidal signals of different frequencies along each of its dimensions (*Figure 5A*).

We defined a simple measure of frequency tuning as the average frequency weighted by sensitivity. We used the *k*-means clustering algorithm to divide the components into two groups (*Figure 5A*,

lower tuning to lower frequencies in red, and to higher in green). The tuning curves of the principal components of the *latent PCs space* are remarkably similar to the ones of the PC and RA afferents (*Figure 5A*, dashed green and red lines, respectively). When we repeated this analysis on the Autoencoder without compression, the tuning profiles were different from the ones of the PC and RA afferents (*Figure 5B*) and from each other (*Figure 5C*); frequency tuning did not cluster as much as for the more compressive 16-feature Autoencoder (*Figure 5C*). Taken together, these results suggest that our sensors for texture (and material) properties have evolved to efficiently encode the statistics of natural textures as they are sensed through vibrations.

## Discussion

We used unsupervised learning to reconstruct the vibratory signals that constitute the very input of the haptic system based on a highly compressed representation. Such representation shares similarities with the perceptual representation of natural materials: it allows for classification of material categories, as provided by human judgments, and crucially, distances between material categories in the latent space resemble perceptual distances. Furthermore, the temporal tuning of the dimensions of the learned representation is similar to properties of human tactile receptors. These similarities suggest that the computations involved in touch perception evolved to efficiently encode the input, yielding perceptual representations that allow to distinguish between different materials. This may be because a way to efficiently represent the vibratory patterns is to discover latent variables that create systematic differences between them, that is, the systematic physical differences between materials.

A similar idea has been proposed for visual perception of material properties (*Fleming and Storrs, 2019*). The challenge for vision is that the information in the retinal image (proximal stimulus) is insufficient to recover the properties of the world (distal stimulus) that caused it (*Anderson, 2011*). In fact, multiple causes are usually responsible for a proximal stimulus, for example, illumination and reflectance spectra are confused within the reflected light. To solve this ambiguity, it is proposed that we learn to represent the dimensions of variation in the proximal stimuli, which arise from the systematic effects of distal stimuli (*Fleming and Storrs, 2019*), rather than learning to directly estimate the distal properties of the world, as predicted by the *inverse optic*s approach (*Marr, 1982*; *Poggio and Koch, 1985*; *Pizlo, 2001*). This approach could successfully be used to predict perception and misperception of gloss (*Storrs et al., 2021*). Our results support the hypothesis that efficiently encoding the proximal stimuli is the way sensory systems develop perceptual representations.

Similar to the ambiguities in the visual input, a challenge for touch perception is that the temporal frequency content of the input vibratory signals depends both on the surface properties and the exploration speed. Nevertheless, we can recognize different materials regardless of the speed of the hand movements used to explore them. This ability of the haptic system can be referred to as speed invariance (*Boundy-Singer et al., 2017*). Our classification analysis could discriminate the material categories even though exploration speed was not controlled. This is consistent with the fact that human roughness judgments can be predicted from nonspeed invariant responses of the PC and RA afferents (*Weber et al., 2013*), presumably because of limited variability in exploration speed. In fact, responses of tactile nerve fibers are highly non-speed invariant, whereas populations of neurons in the somatosensory cortex exhibit different degrees of speed invariance (*Lieber and Bensmaia, 2020*). This may be due to the increased separability of information about texture and speed, which could be implemented by the temporal and the spatial differentiation mechanisms active in the somatosensory cortex (e.g., *Bensmaia et al., 2008*; *Saal et al., 2015*). Speed invariance could also be implemented by normalizing neural responses by movement speed, yielding a representation in spatial coordinates. For this, the haptic system would need a precise and robust measure of the movement speed not only during active but also passive exploration as perceptual speed invariance is found even when textures are passively scanned (*Bensmaia and Hollins, 2003*). There is evidence that an estimate of scanning speed is available (*Dépeault et al., 2008*). An alternative mechanism for speed invariance could be similar to the one mediating (auditory) timbre constancy: while changes in exploration speed affect the frequency composition of the receptors' spiking responses, the harmonic structure remains relatively constant (*Weber et al., 2013*).

It is also possible that observers adjusted their exploration movements to the best speed for texture recognition, yielding a certain level of speed invariance. It is known that exploratory behavior is optimized to maximize information gain (*Lederman and Klatzky, 1987*; *Najemnik and Geisler, 2005*;

*Kaim and Drewing, 2011*; *Toscani et al., 2013*). Specific to exploration speed, it is shown that when participants were asked to haptically discriminate spatial frequencies of gratings, low-performing participants could improve their performance by adjusting the scanning velocity to the one used by better-performing participants (*Gamzu and Ahissar, 2001*). In fact, the statistics of the sensory input are shaped by exploratory behavior, which therefore may contribute to efficient encoding, consistent with the recently proposed 'active efficient coding' theory (*Eckmann et al., 2020*). However, it seems that exploratory strategies depend on the task but not on texture properties. The changes in the properties of exploratory movements (e.g., exploration speed) depending on texture properties probably arise purely from the biophysical interaction with surfaces rather than any changes in motor behavior (*Callier et al., 2015*).

In our study, participants could not access temperature and had limited access to softness information; nevertheless, the perceptual representation inferred from the rating experiment is very similar to the one reported by *Baumgartner et al., 2013*, even though in their study participants could explore materials with their barehands. This suggests that human perception of materials is largely based on vibrations they elicit on the skin. *Bensmaïa and Hollins, 2005* showed that 82% of variance in human dissimilarity judgments between pairs of materials could be explained by the differences in vibrations they elicited on the skin.

Here, we show that efficient encoding of the raw vibratory signals (i.e., measured at the tool with which the surface texture is explored) produces a representation similar to the signals recorded from the afferents responsible for texture perception (i.e., PC and RA). This does not imply that the Autoencoder we tested only mimics the function of the Pacini and Meissner corpuscles innervated by the PC and RA afferents because, before the receptors, the signal might already be compressed by the biomechanic properties of the hand (*Shao et al., 2020*). The mechanosensory representation does not keep all the vibratory information present at the input level: responses of the PC and RA afferents follow a specific tuning profile, which implies selection of vibratory information in the time-frequency domain. Such selection of information can be achieved by learning a compressed representation that gives more weight to some temporal frequency than others. We speculate that the tuning profiles learned by the Autoencoder, as well as the ones exhibited by the PC and RA afferents, are tuned to the statistical properties of vibratory signals elicited by natural material textures during active human exploration.

Crucially, the similarity between perceptual representations and the representations learned by the Autoencoder increases with the level of compression, suggesting a crucial role of efficient encoding of the sensory signals in learning perceptual representations. Because of constraints on perceptual systems, like the number of neurons and the metabolic cost of neural activity (*Lennie, 2003*), receptor input is likely to be compressed and encoded as efficiently as possible to maximize neural resources. Our results suggest that efficient coding forces the brain to discover perceptual representations that relate to the physical factors responsible for variations between signals.

## Materials and methods

### Participants

Eleven students (seven females) participated in the experiment; all were volunteers, naïve to the purpose of the experiment, and were reimbursed for their participation (8 euros/hr). All participants were right-handed and did not report any sensory or motor impairment at the right hand. The study was approved by the local ethics committee LEK FB06 at Giessen University and was in line with the Declaration of Helsinki from 2008. Written informed consent was obtained from each participant.

### Vibratory signals

Participants freely explored 81 out of 84 material samples used by *Baumgartner et al., 2013* with a steel tool tip. Three samples had to be excluded because they were not well preserved. The tip was mounted on a 3D-printed pen and connected to an accelerometer (ADXL345) so that the vibrations elicited by the interaction between the tip and the material samples could be recorded. For each material, 10 s of recording were acquired at 3200 Hz temporal resolution. Then, each recording was parsed into 125 vibratory signals of 80 ms each. This yields a total of 111,375 vibratory signals. To prevent signals from getting affected by the onset and offset of contact between the tool and the

material surface, we started the recording after 2 s of exploration and stopped it 2 s before the exploration was terminated. Frequencies below 10 Hz were ascribed to exploratory hand movements and therefore were filtered out (**Strese et al., 2014**; **Strese et al., 2017**; **Romano and Kuchenbecker, 2014**; **Culbertson et al., 2014**). If not removed, vibrations due to hand movements constitute a potential confound because participants could move their hand differently depending on the material, presumably because of the biophysical interaction between their hand and the surfaces (**Callier et al., 2015**). We also removed frequencies above 800 Hz as they are not relevant for perception of texture properties of materials (**Hollins et al., 2001**; **Bensmala and Hollins, 2003**; **Bensmaïa and Hollins, 2005**; **Weber et al., 2013**; **Manfredi et al., 2014**) and may be caused by measuring noise.

## Rating experiment

Participants sat at a table looking at a monitor on which instructions and rating scales were displayed. Material samples were positioned by the experimenter in front of them. Vision of the samples was prevented by a blind, and sound resulting from the exploration of the materials was covered by earplugs and white noise displayed over headphones. Prior to every trial, participants were informed to grab the pen with the accelerometer and prepare for the exploration. The onset of the white noise signaled that they could begin with the exploration. They were instructed to slide the pen over the material's surface. 2 s after the onset of the white noise, the vibratory signal was recorded for 10 s. After this, participants were informed to stop the exploration. White noise was played for two more seconds. Subsequently, participants reported first how much the explored material felt like each of the seven material categories (paper, fabric, animal, stone, plastic, wood, metal) by clicking for each on a scale ranging from complete dissimilarity to complete similarity with this category (e.g. 'no wood'–'wood'). Then, they rated the material according to seven antagonist adjective pairs again by clicking for each on a scale with the outermost positions corresponding to the two adjectives. We used the same descriptors as used by **Baumgartner et al., 2013**: rough vs. smooth, hard vs. soft, orderly vs. chaotic, warm vs. cold, elastic vs. not elastic, high friction vs. slippery, textured/patterned vs. homogeneous/uniform, omitting descriptors related to the visual appearance (glossiness, colorfulness, three-dimensionality). The experiment was carried out using PsychoPy (**Peirce et al., 2019**). Before all analyses described in the results, we z-transformed the ratings separately for each descriptor and each participant to convert them into a common scale and reduce the impact of subjective criteria.

## Deep neural network

The network used for the analyses described in the article is a relatively simple Deep Convolutional Autoencoder. The *encoder* encodes the 256 time samples of each vibratory pattern into a latent representation consisting of one point in a 16-feature space (i.e., *code*). This is done by means of four 1D convolutional layers, each of them with kernel size equal to five time samples.

The size of the representation in time is progressively reduced by means of a max pooling operation, so that the kernel size is relatively increased for deeper layers. Specifically, after the input layer, the first convolutional layer takes as input signals of 256 time samples and outputs 256 time samples for each of the 64 features. The max pooling operation reduces the time samples to 64. The second convolutional layer takes as input 64 time samples per 64 features and outputs 64 time samples per 32 features. Again, max pooling reduces the time samples; this time to 16. The third convolutional layer takes as input 16 time samples per 32 features and outputs 16 time samples per 16 features. Max pooling reduced the time samples to 4. The fourth convolutional layer takes as input four time samples per 16 features and outputs the same size representation. Finally, max pooling reduces the time samples to one single point for each of the 16 features. This output is the most compressed representation within the network, that is, the *code*, and constitutes the bottleneck of the Autoencoder.

The *code* is decoded by the *decoder* to reconstruct the input signals. The first convolutional layer of the decoder takes as input and outputs one time sample per 16 features. The upsampling operation increases the number of time samples from 1 to 4. This is done by repeating each temporal step four times along the time axis. The second convolutional layer takes as input and outputs four time samples for 16 features. Upsampling increases the time samples from 4 to 16. The third convolutional layer takes as input 16 time samples per 16 features and outputs 16 time samples per 32 features. Upsampling increases the time samples from 16 to 64. The fourth convolutional layer takes as input 64 time samples for 32 features and outputs 64 time samples for 64 features. Upsampling increases

the time samples from 64 to the original size (i.e., 256). The fifth and last convolutional layer takes as input 256 time samples per 64 features and outputs 256 time samples. Thus, input and output have the same size and can be directly compared within the loss function.

The activation function (∏) of all convolutional layers is the rectified linear unit (*Equation 2*), with the exception of the last layer of the decoder, for which it is a sigmoid function (*Equation 3*).

$$\Pi\left(x\right) = max\left(0, x\right) \tag{2}$$

$$\Pi\left(x\right) = 1/\left(1 + e^{-x}\right) \tag{3}$$

Training consisted of 50 epochs to minimize the *mean absolute error (MAE)* loss function:

$$MAE = \frac{1}{N}\sum_{i=0}^{N}\left|y_i - \hat{y}_i\right| \tag{4}$$

With $y_i$ the $i$ vibratory signal of the training set (consisting of N signals), and $y_i$ the corresponding reconstructed signal.

The network was adapted from *Chollet, 2016* to work with one-dimensional data rather than images. The implementation and training of the network were done through the Keras deep learning framework (*Chollet, 2015*) via its Python interface.

## Networks with different compression rates
We started from the architecture described above and changed the bottleneck to reduce the level of compression. To do so, we increased the number of features in the latent space (i.e., from 16 to 32, 64, 192, 256; with compression rate from 6.25% to 25, 50, 75, and 100%). To make sure that none of the intermediate layers had a lower dimensionality than the latent space, we imposed their number of features to be not less than the dimensionality of the latent space.

## Linear classification
Classification was done based on the *latent PCs space* dimensions by iteratively leaving out one material per category, training the classifier on the remaining signals, and computing performance on the leftout samples. Before classification, the latent representations of all signals of each material sample were averaged, so that there was one point for each material sample. As a conservative method for hypothesis testing, we used bootstrap analysis to assess the significance of all classification results. For that, we repeated the classification analysis 5000 times under the null hypothesis of no relationship between the points in the *latent PC space* and the material categories, that is, we shuffled the category labels. This produced the accuracy distribution under the null hypothesis of chance-level classification; the 95% confidence interval was computed by reading out the 2.5th and 97.5th percentiles of the distribution. The empirical chance level corresponded to the mean of that distribution. The labels used for training and testing were either the ground truth material categories (example in *Figure 3B*) or the ones assigned by participants (example in *Figure 3C*). To assign category labels to different material samples, first we assigned to each material the highest rated material category by each participant, then we chose the most frequent category, that is, the one chosen by the majority of participants. The between-participants agreement level is computed based on each participant's category rating. Again, for each sample, we assigned a category label as the highest rated category for each participant. Then, we considered each participant for testing, iteratively, and computed the predicted category as the one chosen by the majority of the other participants. The agreement level is the classification accuracy computed by comparing the predicted and the testing category labels.

## Temporal tuning
We fed the network with 830 sinusoids with frequencies ranging from 0 to 800 Hz and computed their representation in the *latent PCs space*. Temporal tuning is given by the normalized (range is forced between 0 and 1) model responses as a function of temporal frequency. For the tactile afferents, sensitivity is computed as normalized inverse of the discrimination thresholds from *Kandel et al., 2000*; *Mountcastle et al., 1972*.

## Data availability

The analysis code is publicly available. The code for processing the vibratory signals and the Python code for training the DNNs and saving the reconstructed signals and the latent representations is available at GitHub (here; *Toscani, 2021* copy archived at swh:1:rev:53b1d7407307c00f08543cad096f983217a53ef2).

## Acknowledgements

This work was supported by the Deutsche Forschungsgemeinschaf (DFG, German Research Foundation) – project number 222641018 – SFB/TRR 135. We are grateful to Karl Gegenfurtner and Knut Drewing for helpful discussions and comments.

## Additional information

### Funding

| Funder | Grant reference number | Author |
| --- | --- | --- |
| Deutsche Forschungsgemeinschaft | 222641018 - SFB/TRR 135 | Matteo Toscani Anna Metzger |

The funders had no role in study design, data collection and interpretation, or the decision to submit the work for publication.

### Author contributions

Anna Metzger, Matteo Toscani, Conceptualization, Data curation, Formal analysis, Methodology, Software, Validation, Visualization, Writing – original draft, Writing – review and editing

### Author ORCIDs

Anna Metzger http://orcid.org/0000-0002-5704-2821
Matteo Toscani http://orcid.org/0000-0002-1884-5533

### Ethics

The study was approved by the local ethics committee LEK FB06 at Giessen University and was in line with the declaration of Helsinki from 2008. Written informed consent was obtained from each participant.

### Decision letter and Author response

Decision letter https://doi.org/10.7554/eLife.64876.sa1
Author response https://doi.org/10.7554/eLife.64876.sa2

## Additional files

### Supplementary files

• Transparent reporting form

### Data availability

All data are publicly available online (https://github.com/matteo-toscani-24-01-1985/Unsupervised-learning-of-haptic-material-properties copy archived at swh:1:rev:53b1d7407307c00f08543cad096f983217a53ef2).

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
