## [Editor Report]

The authors apply machine learning techniques to empirical-based simulations of the vibrotactile patterns to find their compressed representations. These compressed representations turn out to be similar to those deduced from human responses and also correspond to those expected based on efficiency arguments.

---

## [Decision Letter]

**Decision letter after peer review:**

Thank you for submitting your article "Unsupervised learning of haptic material properties" for consideration by *eLife*. Your article has been reviewed by 2 peer reviewers, and the evaluation has been overseen by a Reviewing Editor and Chris Baker as the Senior Editor. The following individuals involved in review of your submission have agreed to reveal their identity: Ehud Ahissar (Reviewer #1); Sliman J Bensmaia (Reviewer #2).

The reviewers have discussed the reviews with one another and the Reviewing Editor has drafted this decision to help you prepare a revised submission. The main concern is about classification performance. It should be demonstrated that the present approach provides substantial improvements compared to more standard and simpler methods. However, the reviewers have expressed doubt that this is the case during the discussion stage. However, we would like to give you an opportunity to address this and other comments and therefore are inviting a revised manuscript. If you do not think that this can be addressed, then it is unlikely that the manuscript will ultimately be accepted for publication.*Reviewer #1:*

The paper describes a very interesting attempt to understand texture coding by touch using autoencoding (AE) of empirical-based simulations of the vibrotactile patterns that would dominate receptor activation during tactile scanning of various surfaces. By showing that the compressed representation generated by the AE allows surface classification, with distances that are correlated to some degree with those exhibited by human perceivers, the authors suggest that the latent representations underlying human perception might be similar to that observed in the simulations. The authors further claim that the simulated latent representation is efficient, and that it matches the characteristics of RA and PC mechanoreceptors. These observations lead the authors to suggest that haptic representation is dictated by efficient coding.

1. There is a significant gap between the simulated data used here and the empirical data of material perception by touch. The vibratory signals were taken from recordings of surface exploration using a tool tip (Strese et al., 2017) whereas the ratings of the different materials are taken from an experiment in which participants used bare hand touch (Baumgartner et al., 2013). The difference is significant especially when material classification, and not only texture classification, is required. It is not at all clear how vibratory signals could code hardness, warmth, elasticity, friction, 3D, etc (see Baumgartner et al., 2013). The authors must provide a serious discussion about this gap and convince the reader that their simulations can indeed provide an access to the internal representations of natural haptic touch. In the same spirit, they should explain why, and demonstrate that, the pre-processing of the vibrotactile data (cutting and filtering) makes sense for natural haptic touch.

2. The authors should provide good reasons to convince the readers that the compressed representation they found is indeed a good candidate for the biological representation. First, the nature of the AE algorithm is that it will converge to some representation in the minimal encoder dimension. Why is that a good encoding representation, and why is it a good model for the biological one? Second, the differences between the results obtained with the AE and those obtained with humans (Baumgartner et al., 2013) seem to outnumber the partial similarities found. The authors should list both differences and similarities and discuss, based on these comparisons, the probability that the coding found here is similar to the coding guiding human behavior.

3. Notion of efficiency and compression. It was not demonstrated that the main result (figure 3A) is due to compression and efficiency of the AE. What will happen if no AE is used and the distance is measured in the raw input domain (e.g. between Fourier coefficients or principal components)? One could expect figure 4 to account for that, and also show that for a very wide AE there is some deterioration of the main result. Otherwise, the main result about correlation to perceptual data cannot be attributed to the compressive property of the AE.

4. Biological correlates of the latent representation. On the one hand the authors claim that the AE latent representation aims to mimic a latent representation of the haptic space, which they assume to be compressed and efficient. On the other hand, they claim that the AE representation is similar to mechano-sensory representation, which is a first biological representation before any compression can take place (when hand movements are ignored, as done here). This needs to be clarified.

5. Validity of the latent representation. The reconstruction error of the AE is large and systematic: only ~50% of the variance are explained and its high frequencies are systematically ignored. The resulting latent representation is such that classes are poorly separable (~29%) and it seems to be by far worse than the human level (around 70% in Baumgartner et al., 2013). It will be therefore interesting to see if the key result, i.e. relation between AE latent space and the perceptual distance, remains valid for a more advanced AE.

*Reviewer #2:*

The goal of using a deep neural network to understand how neuronal representations of tactile texture are constructed is exciting and potentially promising. My enthusiasm for this paper is diminished by the poor performance of the classification and the weak relationship between latent space and perceptual ratings. Indeed, the output of the autoencoder preserves texture information only at a very coarse granularity, resulting in poor classification and poor perceptual predictions.

The main problem is that the latent factors yield poor classification performance and are only weakly related to perceptual judgments. Indeed, analogous analyses (without fancy machine learning algorithms) tend to perform better on both fronts (classification and perceptual prediction).

Another issue is that the paper lacks focus: the modest results are cast in the context of a long and somewhat pedantic discussion of optimality.

The discussion of texture invariance omits two important threads. First, exploratory procedures have been shown not to be optimized on a texture by texture basis (Callier et al., JNP, 2015), as is suggested could be the case. Second, the neural mechanisms that mediate invariance have been discussed (Lieber et al., Cerebral Cortex, 2020) beyond speculations about timbre (Yau et al., Communicative and Integrative Biology, 2009).

The comparison of the performance of different machine learning approaches does not seem to yield any additional insight and is probably better relegated to supplemental materials.

---

## [Author Response]

Reviewer #1:The paper describes a very interesting attempt to understand texture coding by touch using autoencoding (AE) of empirical-based simulations of the vibrotactile patterns that would dominate receptor activation during tactile scanning of various surfaces. By showing that the compressed representation generated by the AE allows surface classification, with distances that are correlated to some degree with those exhibited by human perceivers, the authors suggest that the latent representations underlying human perception might be similar to that observed in the simulations. The authors further claim that the simulated latent representation is efficient, and that it matches the characteristics of RA and PC mechanoreceptors. These observations lead the authors to suggest that haptic representation is dictated by efficient coding.1. There is a significant gap between the simulated data used here and the empirical data of material perception by touch. The vibratory signals were taken from recordings of surface exploration using a tool tip (Strese et al., 2017 ) whereas the ratings of the different materials are taken from an experiment in which participants used bare hand touch (Baumgartner et al., 2013). The difference is significant especially when material classification, and not only texture classification, is required. It is not at all clear how vibratory signals could code hardness, warmth, elasticity, friction, 3D, etc (see Baumgartner et al., 2013). The authors must provide a serious discussion about this gap and convince the reader that their simulations can indeed provide an access to the internal representations of natural haptic touch.

We agree with the reviewer that perception of materials is not exclusively limited to vibrations. Indeed, a review of studies investigating the psychophysical dimensions in texture perception revealed five potential dimensions: macro and fine roughness, warmness/coldness, hardness/softness, and friction (Okamoto, Nagano and Yamada, 2013).

We collected new behavioral data, based on explorations with a tool tip, in which participants had no or very limited access to temperature or softness information and mainly had to rely on the sensed vibratory signals. We compared the perceptual representation inferred from our data with the one from the Baumgartner at al. study (2013) and found that these representations are remarkably similar. Thus we are confident that the missing temperature and softness information did not much affect the perceptual representations we inferred from the rating experiments, which could therefore be thought as representative for natural haptic touch.

This is explained in the “Behavior” paragraph we added to the Results, and discussed in the discussion of the revised version:

“In our study participants could not access temperature and had limited access to softness information, nevertheless the perceptual representation inferred from the rating experiment are very similar to the one reported by Baumgartner and colleagues (2013), even though in their study participants could explore materials with their bare hands. This suggests that human perception of materials is largely based on vibrations they elicit on the skin. Bensmaïa and Hollins (2005) showed that 82% of variance in human dissimilarity judgments between pairs of materials could be explained by the differences in vibrations they elicited on the skin.”

Our behavioral experiment (similar to the one form Baumgartner and colleagues) is explained in the “Rating experiment” section in the methods of the revised version.

In the same spirit, they should explain why, and demonstrate that, the pre-processing of the vibrotactile data (cutting and filtering) makes sense for natural haptic touch.

The reason why we cut the beginning of samples was that we wanted to exclude the vibrations elicited by the contact of the tool tip with the surfaces. For the new measures we started to record after two seconds each trial begun and stopped two seconds before each trial was over. Thus cutting was no longer necessary. This is explained in the methods:

“To prevent that signals are affected by the onset and offset of contact between the tool and the material surface we started the recording after 2 seconds of exploration and stopped it 2 seconds before the exploration was terminated”

We filtered our data to focus on the relevant frequency range, thus reducing complexity and potentially also reducing noise and potential confounds. Filtering out frequencies due to exploratory hand movements is a common approach (e.g. Strese et al., 2014, Strese et al., 2017; Romano and Kuchenbecker, 2014; Culbertson, Delgado and Kuchenbecker, 2014) and it controls for a possible confound, i.e. that people explored the textures with a different dynamics depending on the materials, so that the frequencies due to exploratory hand movements would be diagnostic for material differences. For this, we followed Strese et al., (2014) and removed frequencies below 10 Hz. We write this in the methods:

“Frequencies below 10Hz were ascribed to exploratory hand movements and therefore were filtered out (e.g. Strese et al., 2014, Strese et al., 2017; Romano and Kuchenbecker, 2014; Culbertson, Delgado and Kuchenbecker, 2014). If not removed, vibrations due to hand movements constitute a potential confound because participants could move their hand differently depending on the material, presumably because of the biophysical interaction between their hand and the surfaces (Callier et al., 2015).”

We also removed frequencies above 800Hz because they are scarcely relevant for perception of texture, as they exceed the limit of the sensitivity of the Pacinian receptors. We comment on this in the methods:

“We also removed frequencies above 800Hz, as they are not relevant for perception of texture properties of materials (Hollins, Bensmaïa and Washburn, 2001; Bensmaïa and Hollins, 2003; Bensmaïa and Hollins, 2005; Weber et al., 2013; Manfredi et al., 2014) and may be caused by measuring noise.”

2. The authors should provide good reasons to convince the readers that the compressed representation they found is indeed a good candidate for the biological representation. First, the nature of the AE algorithm is that it will converge to some representation in the minimal encoder dimension. Why is that a good encoding representation, and why is it a good model for the biological one?

We do not claim that the AE algorithm is the best model for efficient encoding. We argue that the AE is a plausible model for it because it needs to learn a compressed representation of the sensory input and by doing so it may discover the latent variables caused by reoccurring physical properties of the stimuli, e.g. general properties of wood, metal, plastic, etc. We write this in the discussion of the revised version.

“Because of the bottleneck Autoencoders must learn a compressed representation from which the input signals can still be reconstructed. To do so, they tend to discover latent variables that are good at capturing the pattern of covariations in the input generated by distal causes (e.g. material properties of the surfaces, Fleming and Storrs, 2019; Storrs and Fleming, 2021).

Our point is that by compressing the sensory input we learn general patterns of covariation which may reflect the physical properties of the world. In fact, with the new analysis we present in the revised version, we show that the similarity between perceptual representations and the representation learned by the AE depends on the compression level:

“Our results showed a remarkable similarity between perceptual representations and how material categories are represented in the latent space learned by the Autoencoder. This is consistent with the theory that perceptual representations emerge from efficient encoding of the sensory signals. Thus, when encoding is less efficient, the similarity between learned and perceptual representations should be reduced. We tested this prediction by gradually decreasing the compression rate of the Autoencoder until no compression is applied, i.e. the dimensionality of the latent space is the same as the one of the input. Higher compression rate, defined as the ration between the dimensionality of the latent space and the one of the input (in percentage), implies lower compression and results in higher reconstruction performance (Figure 4D, blue continuous and dashed lines, for the validation and the training data-set, respectively). The least compression (higher compression rate) the lower the correlation between perceptual and latent space distance matrices (Figure 4D – black) and the lower the classification performance (Figure 4D – orange). These results show that compression is crucial for the Autoencoder to learn an internal representation that resembles perceptual representations.”

We discuss the general theory in the discussion:

“A similar idea has been proposed for visual perception of material properties (Fleming and Storrs, 2019). The challenge for vision is that the information in the retinal image (proximal stimulus) is insufficient to recover the properties of the world (distal stimulus) that caused it (Anderson, 2011). In fact, multiple causes are usually responsible for a proximal stimulus, e.g. illumination and reflectance spectra are confused within the reflected light. To solve this ambiguity, it is proposed that we learn to represent the dimensions of variation in the proximal stimuli, which arise from the systematic effects of distal stimuli (Fleming and Storrs, 2019), rather than learning to directly estimate the distal properties of the world, as predicted by the inverse optics approach (Marr, 1982; Poggio and Koch, 1985; Pizlo, 2001). This approach could successfully be used to predict perception and misperception of gloss (Storrs, Anderson and Fleming, 2021). Our results support the hypothesis that efficiently encoding the proximal stimuli is the way sensory systems develop perceptual representations.”

Second, the differences between the results obtained with the AE and those obtained with humans (Baumgartner et al., 2013) seem to outnumber the partial similarities found. The authors should list both differences and similarities and discuss, based on these comparisons, the probability that the coding found here is similar to the coding guiding human behavior.

We believe that the major reason for the differences we found between the results we obtained with the AE and those obtained with humans is that we tested the AE on vibrations elicited by a different set of stimuli than the ones used for the behavioral data in Baumgartner et al., (2013). Given this, we were actually surprised to find similarities at all, and we found our results striking. However, ideally one would collect rating judgments and vibratory signals with the same material samples and exploratory movements, and under the same conditions (i.e. no temperature and limited softness signals available). We did this for the revised version: we build our measuring setup and borrowed the stimuli used by Baumgartner and colleagues. This also allowed a direct comparison with their data, which revealed that perceptual representations inferred from exploration with the tool tip are very similar to the ones from bare hand exploration, i.e. our perceptual data are likely to be ecologically valid.

Analysis on the new data show remarkable similarities between results we obtained with the AE and those obtained with humans. The correlation between the distance matrix (indicating the structure of perceptual representations) between the category averages in the latent space of the Autoencoder with the one obtained from human data is remarkably high and much higher than in the original version (r=0.74 vs r=0.55). Classification performance improved when using labels assigned by human participants, and it is overall higher than in the previous version, and more importantly, close to the between-observers agreement level. This suggests that the model is able to capture nearly all the information available for the perceptual judgments expressed by our participants.

3. Notion of efficiency and compression. It was not demonstrated that the main result (figure 3A) is due to compression and efficiency of the AE. What will happen if no AE is used and the distance is measured in the raw input domain (e.g. between Fourier coefficients or principal components)?

We thank the reviewer for this important insight. In the original version we missed the opportunity to directly test our theory that perceptual representations emerge from efficient encoding (which implies compression). However, the test is straightforward, as the compression level can be systematically varied by changing the bottleneck of the Autoencoder. We repeated all analysis at different compression levels and found that the similarity between perceptual representations and the representation in the latent space increases with the compression level.

These new results are presented in the “results” section:

“Our results showed a remarkable similarity between perceptual representations and how material categories are represented in the latent space learned by the Autoencoder. […] To do so, they tend to discover latent variables that are good at capturing the pattern of covariation in the input generated by distal causes (e.g. material properties of the surfaces, Fleming and Storrs, 2019; Storrs and Fleming, 2021).”

We also followed the suggestion of repeating our analysis with a simple linear compression algorithm (i.e. PCA) rather than the AE. With decreasing compression level, i.e. increasing dimensionality of the bottleneck (16, 64, 128, 192 and 256 Principal Components) classification performance decreases, like for the AE, but it is generally much lower than the one obtained with the AE and at chance level for lower compression levels (24%, 25%, 14%, 14% and 14%, for the bottleneck dimensionalities listed above). The correlation between the perceptual between-category distances and the between-category distances in the latent space is rather independent of the compression level, and never statistically different from zero (rs = 0.3455, 0.3490, 0.3490, 0.3488, 0.3488). The AE seems to be a better model of efficient encoding, as suggested by Fleming and Storrs (2019), although probably other algorithms may learn perceptual representations by compressing the input signals. Note that the mile stone paper by Buchsbaum and Gottschalk (1983) on efficient encoding of color information is based on a principal component transformation.

One could expect figure 4 to account for that, and also show that for a very wide AE there is some deterioration of the main result. Otherwise, the main result about correlation to perceptual data cannot be attributed to the compressive property of the AE.

We thank again the review for pointing this out as we failed to directly test the effect of compression on our results. In Figure 4 of the original version the maximum compression rate we tested was 12%. In the new analysis we go from 6.25% to 100%. By inspecting our results in Figure 4 of the revised version it appears that no much deterioration is to expect around 12% as things start to change with higher compression rates.

4. Biological correlates of the latent representation. On the one hand the authors claim that the AE latent representation aims to mimic a latent representation of the haptic space, which they assume to be compressed and efficient. On the other hand, they claim that the AE representation is similar to mechano-sensory representation, which is a first biological representation before any compression can take place (when hand movements are ignored, as done here). This needs to be clarified.

We are not sure we understand the criticism here. The mechano-sensory representation does not keep all the vibratory information as it can be roughly approximated by a filter in the frequency domain. Such selection of information can be achieved by learning a compressed representation that keeps information at some temporal frequency and throws others out. Similarly, chromatic signals are decorrelated by the retinal circuitry to efficiently transmit them to the brain. Such efficient encoding happens at the very input stage of the visual system and implies color-opponency, which is one of the primary properties of how color is represented perceptually.

We try to spell this out in the discussion of the revised version.

“The mechano-sensory representation does not keep all the vibratory information present at the input level: responses of the PC and RA afferents follow a specific tuning profile which implies selection of vibratory information in the time frequency domain. Such selection of information can be achieved by learning a compressed representation that gives more weight to some temporal frequency than others. We speculate that the tuning profiles learned by the Autoencoder, as well as the ones exhibited by the PC and RA afferents, are tuned to the statistical properties of vibratory signals elicited by natural material textures during active human exploration.”

5. Validity of the latent representation. The reconstruction error of the AE is large and systematic: only ~50% of the variance are explained and its high frequencies are systematically ignored. The resulting latent representation is such that classes are poorly separable (~29%) and it seems to be by far worse than the human level (around 70% in Baumgartner et al., 2013). It will be therefore interesting to see if the key result, i.e. relation between AE latent space and the perceptual distance, remains valid for a more advanced AE.

Our idea is that the AE learns what is perceptually relevant and discards the rest, because it represents a plausible model of how information is efficiently encoded by biological systems. So, we are not worried about the proportion of explained variance. Actually, higher compression (thus less explained variance) implies higher similarities with perceptual representations.

With respect of the poor classification accuracy, the new analysis revealed that our AE model performs very close to the between-participants agreement which gives a reasonable measure of upper performance limit. The correlation between AE latent space and the perceptual distance is also high (r=0.74) indicating that representations are remarkably similar and there is no need for a more complex AE.

We believe that the mismatch between perceptual representations and the representations learned by the AE reported in the original version of the manuscript mainly arose because for deep learning we used vibratory signals elicited by different stimuli than the ones used for the behavior experiments. After solving this problem, we think that our manuscript has significantly improved.

Reviewer #2:The goal of using a deep neural network to understand how neuronal representations of tactile texture are constructed is exciting and potentially promising. My enthusiasm for this paper is diminished by the poor performance of the classification and the weak relationship between latent space and perceptual ratings. Indeed, the output of the autoencoder preserves texture information only at a very coarse granularity, resulting in poor classification and poor perceptual predictions.

We believe that the major reason for the poor classification performance and the weak relationship between the latent space and the perceptual ratings is that we tested the AE on vibrations elicited by a different set of stimuli than the ones used for the behavioral data in Baumgartner et al., (2013). Our idea is that by efficient encoding the AE and biological systems discover the latent variables which caused the signal and by this e.g. typical vibratory patterns for different material categories. So in principle, perceptual and latent spaces should be similar independent of the specific material samples used for training the AE and in the behavioral experiment. However, this requires a huge number of samples. For instance, Storrs and Fleming (2021, Nature) used 10 000 images to train the AE. Available databases of vibratory recordings are 100 times smaller. Given this, we were actually surprised to find similarities between the latent and the perceptual space based on different material samples at all, and we found our results striking. However, ideally, one would collect rating judgments and vibratory signals with the same material samples and exploratory movements, and under the same conditions (i.e. no temperature and limited softness signals available). We did this for the revised version: we build our measuring setup and borrowed the stimuli used by Baumgartner and colleagues. This also allowed a direct comparison with their data, which revealed that perceptual representations inferred from exploration with the tool tip are very similar to the ones from bare hand exploration, i.e. our perceptual data are likely to be ecologically valid.

Analysis on the new data show remarkable similarities between results we obtained with the AE and those obtained with humans. The correlation between the distance matrix (indicating the structure of perceptual representations) between the category averages in the latent space of the AE with the one obtained from human data is remarkably high and much higher than in the original version (r=0.74 vs. r=0.55). The classification performance also improved as compared to the original version (36% vs 29%)and it is even higher when using labels assigned by human participants (40%). More importantly it is close to the between-observers agreement level (44%). This suggests that the model is able to capture nearly all the information available for the perceptual judgments expressed by our participants.

We hope our new data an analysis could restore the reviewer‘s enthusiasm as much as it boosted ours.

The main problem is that the latent factors yield poor classification performance and are only weakly related to perceptual judgments. Indeed, analogous analyses (without fancy machine learning algorithms) tend to perform better on both fronts (classification and perceptual prediction).

With the new data and analysis we show that the latent factors yield classification performance nearly as high as between-participants agreement and that they are highly correlated with perceptual judgments. However, we thank the reviewer for the opportunity to explain ourselves better with respect of the choice of the algorithm. We wanted to test the hypothesis that perceptual representations emerge by compressing sensory information to learn to reconstruct the ambiguous sensory input, with no access to the physical ground-truth. This is what autoencoders do, that‘s why we followed the theory of Fleming and Storrs (2019) and used an autoencoder. Note that our autoencoder is much simpler than the one used by Storrs and Fleming (Nature, 2021) to learn visual representation of complex material properties (like gloss). In fact, by systematically varying compression level in the new anaylsis, we discover that compression determines the similarity between perceptual representations and the representations emerged in the latent space. This is in line with the theory that a compressive bottleneck is necessary to discover latent variables that are good at capturing the pattern of covariation in the input generated by distal causes (e.g. material properties of the surfaces) (Storrs and Fleming, 2019; 2021).

We comment on this while presenting the new results about the compression level in the revised version:

“Our results showed a remarkable similarity between perceptual representations and how material categories are represented in the latent space learned by the Autoencoder. […] To do so, they tend to discover latent variables that are good at capturing the pattern of covariation in the input generated by distal causes (e.g. material properties of the surfaces, Fleming and Storrs, 2019; Storrs and Fleming, 2021).”

The theory is also presented and related to the sense of vision, for which it was proposed, in the discussion

A similar idea has been proposed for visual perception of material properties (Fleming and Storrs, 2019). The challenge for vision is that the information in the retinal image (proximal stimulus) is insufficient to recover the properties of the world (distal stimulus) that caused it (Anderson, 2011). In fact, multiple causes are usually responsible for a proximal stimulus, e.g. illumination and reflectance spectra are confused within the reflected light. To solve this ambiguity, it is proposed that we learn to represent the dimensions of variation in the proximal stimuli, which arise from the systematic effects of distal stimuli (Fleming and Storrs, 2019), rather than learning to directly estimate the distal properties of the world, as predicted by the inverse optics approach (Marr, 1982; Poggio and Koch, 1985; Pizlo, 2001). This approach could successfully be used to predict perception and misperception of gloss (Storrs, Anderson and Fleming, 2021). Our results support the hypothesis that efficiently encoding the proximal stimuli is the way sensory systems develop perceptual representations.

Another issue is that the paper lacks focus: the modest results are cast in the context of a long and somewhat pedantic discussion of optimality.

We thank the reviewer for giving us the chance to clarify: rather than a general theory of optimal encoding of information, we focus on compression and unsupervised learning, starting from the theory of Fleming and Storrs (2019). We hope this is clearer in the revised version.

The discussion of texture invariance omits two important threads. First, exploratory procedures have been shown not to be optimized on a texture by texture basis (Callier et al., JNP, 2015), as is suggested could be the case.

We thank the reviewer for pointing out their results. We included this in the discussion in the revised version:

“although exploration movements may be optimized for information sampling depending on the task, it seems that they are not optimized on a texture by texture basis”

Second, the neural mechanisms that mediate invariance have been discussed (Lieber et al., Cerebral Cortex, 2020) beyond speculations about timbre (Yau et al., Communicative and Integrative Biology, 2009).

We agree with the reviewer that our discussion may be partial and misleading. Speed-invariance is of course an important topic, but out of the focus of our study. We only wanted to state that we do not claim that the representation learned by the autoencoder is speed-invariant, as we think that it roughly corresponds to the output of the PC and the RA afferents. Nevertheless, the representation emerged in the latent space allows for classification of material categories (as well as the non-speed invariant responses of the PC and RA afferents), presumably because the variability caused by differences in the exploration speed is less than the variability between categories.

Although speed-invariance wasn’t our focus, we agree we addressed the topic superficially and thank the reviewer for their criticism.

We changed the paragraphs about optimal exploration and speed-invariance to be clearer and more exhaustive:

“Similar to the ambiguities in the visual input, a challenge for touch perception is that the temporal frequency content of the input vibratory signals depends both on the surface properties and the exploration speed. […] The changes in the properties of exploratory movements (e.g. exploration speed) depending on texture properties probably arise purely from the biophysical interaction with surfaces rather than any changes in motor behaviour (Callier, Saal, Davis-Berg and Bensmaia, 2015).”

The comparison of the performance of different machine learning approaches does not seem to yield any additional insight and is probably better relegated to supplemental materials.

We agree with the reviewer that a non-systematic exploration of different machine learning approaches does not yield additional insights. By following the suggestions received during the reviewing process we think we understood that the compression level imposed by the bottleneck is the crucial factor to determine the similarity between the latent representation and perceptual representations. We removed the comparisons analysis from the revised version and only report the effects of different bottlenecks.